# The Biological and Clinical Relevance of G Protein-Coupled Receptors to the Outcomes of Hematopoietic Stem Cell Transplantation: A Systematized Review

**DOI:** 10.3390/ijms20163889

**Published:** 2019-08-09

**Authors:** Hadrien Golay, Simona Jurkovic Mlakar, Vid Mlakar, Tiago Nava, Marc Ansari

**Affiliations:** 1Platform of Pediatric Onco-Hematology research (CANSEARCH Laboratory), Department of Pediatrics, Gynecology, and Obstetrics, University of Geneva, Bâtiment La Tulipe, Avenue de la Roseraie 64, 1205 Geneva, Switzerland; 2Department of Women-Children-Adolescents, Division of General Pediatrics, Pediatric Onco-Hematology Unit, Geneva University Hospitals (HUG), Avenue de la Roseraie 64, 1205 Geneva, Switzerland

**Keywords:** G protein-coupled receptor (GPCR), hematopoietic stem cell transplantation, treatment-related toxicities, mobilization, engraftment, plerixafor

## Abstract

Hematopoietic stem cell transplantation (HSCT) remains the only curative treatment for several malignant and non-malignant diseases at the cost of serious treatment-related toxicities (TRTs). Recent research on extending the benefits of HSCT to more patients and indications has focused on limiting TRTs and improving immunological effects following proper mobilization and engraftment. Increasing numbers of studies report associations between HSCT outcomes and the expression or the manipulation of G protein-coupled receptors (GPCRs). This large family of cell surface receptors is involved in various human diseases. With ever-better knowledge of their crystal structures and signaling dynamics, GPCRs are already the targets for one third of the current therapeutic arsenal. The present paper assesses the current status of animal and human research on GPCRs in the context of selected HSCT outcomes via a systematized survey and analysis of the literature.

## 1. Introduction 

### 1.1. Hematopoietic Stem Cell Transplantation (HSCT) 

The field of hematopoietic stem cell transplantation (HSCT) has witnessed t remendous progress since its origins in the 1950s [1]#. The number of HSCTs has exploded, along with its range of indications, candidates, and donor sources [2,3]#. HSCT remains indispensable for treating several malignant and non-malignant disorders. The use of peripheral blood stem cells (PBSCs) is well established in autologous transplantation [4]#, and they have become the preferred source of allogeneic hematopoietic stem cells (HSCs), at least in adults [5,6]#. In both scenarios, the number of circulating HSCs mobilized from the bone marrow is closely associated with the engraftment outcome [7]#. Before the graft infusion, most HSCT protocols require a preparation phase, which aims to kill malignant cells to make room for the newly infused HSCs to engraft or to induce immunosuppression. The latter is important to avoid graft rejection and graft-versus-host disease (GvHD) in allogeneic settings. This so-called conditioning regimen comprises high doses of chemotherapeutic drugs and/or radiotherapy that cause a cytotoxic burst of tumor and/or normal cells. This results in a pro-inflammatory status [8,9]#, which is desired when treating malignant conditions with allogeneic HSCT, as it promotes a graft-versus-leukemia (GvL) effect. On the other hand, uncontrolled inflammation results in serious treatment-related toxicities (TRTs), such as sinusoidal obstruction syndrome (SOS), lung toxicity, or GvHD, the negative counterpart of GvL. Recent research efforts have focused on limiting transplantation- or treatment-related mortality (TRM) and TRTs while improving the beneficial immunological effects after adequate mobilization and engraftment.

### 1.2. G Protein-Coupled Receptors (GPCRs)

Human cells express some 400 non-olfactory G protein-coupled receptors (GPCRs) [10]# that respond to a large variety of ligands and thereby affect key cellular functions such as survival or proliferation [11]#. GPCRs, also known as seven-transmembrane spanning receptors (7TMRs), are classified into five subfamilies based on sequence homology: glutamate, rhodopsin, adhesion, frizzled/taste2, and secretin [12]#. Mechanistically, a ligand binding to its cognate GPCR induces the dissociation and the activation of α and βγ subunits in the associated G protein. This signaling ceases upon GPCR phosphorylation by a GPCR kinase (GRK), which causes the binding of a β-arrestin and GPCR endocytosis [13]#. Regulation occurs at multiple levels, whereas GRKs and β-arrestins can also generate G protein-independent GPCR signaling [14,15,16,17]#. GPCRs or their ligands are targeted by a third of all approved drugs, and many more are in development [18]#. Their ubiquity and location at the cell surface make them attractive targets. In addition, many GPCR-ligand crystal structures are already available, and steady progress in the fields of crystallization and molecular modeling has facilitated GPCR drug development [19,20]#. For instance, the discovery of “biased” GPCR agonists, which can differentially activate G protein-dependent and G protein-independent signaling, holds the promise of fine-tuning the pharmacological modulation of GPCRs [21,22]#. Importantly, GPCRs have been linked to multiple human diseases [23,24,25,26]#, and their roles in regulating inflammation are increasingly recognized. Lipid mediators produced by innate immune cells such as the eicosanoids, which include prostaglandins, thromboxanes, and leukotrienes, signal via GPCRs to initiate acute inflammation [27]#. Other related classes of GPCR lipid ligands, such as endocannabinoids [28]#, sphingolipids [29]#, or even the so-called specialized pro-resolving mediators (SPMs) [30]#, also participate in the regulation of inflammation. In turn, chemokines can activate, via GPCRs, both innate and adaptive immune cells and regulate their traffic between lymphoid organs and inflammatory sites [31]#. Finally, although the Beta-2 adrenergic receptor (B2AR) was the first GPCR ever cloned [32]#, the role of adrenergic receptors in modulating immunity and inflammation has only recently been brought to light [33,34,35]#. Discussing time and tissue-specific expression patterns of each GPCR is beyond the scope of this review, but this type of information can be found in part in the references above or in public databases such as the Human Protein Atlas (HPA) or the Genotype-Tissue Expression (GTEx) Portal.

### 1.3. HSCT and GPCR: Plerixafor and Beyond

C-X-C receptor 4 (CXCR4) is a noticeable example of the convergence between GPCRs and HSCT. C-X-C ligand 12 (CXCL12) is a chemokine produced by the stromal cells populating the hematopoietic niche in the bone marrow (BM) [36]#. CXCL12 exerts its function by binding CXCR4, a GPCR present on the surface of hematopoietic cells. That interaction is essential for the homing and the maintenance of HSCs in the BM [37]#. In 2008, the Food and Drug Administration (FDA) approved the use of the first-in-class CXCR4-inhibitor, plerixafor (AMD3100; Mozobil^®^), in association with granulocyte colony-stimulating factor (G-CSF) for the mobilization and the collection of PBSC in patients with non-Hodgkin’s lymphoma (NHL) and multiple myeloma (MM) [38,39,40]#. In this regard, the present paper assesses the new developments concerning plerixafor, which have occurred since the last systematic review on the topic [41]#. Importantly, this paper also surveys the available evidence linking other GPCRs to HSCT outcomes, for which there was not any existing comprehensive review (see the [App secA1-ijms-20-03889]). With this in mind, we prepared a systematized search of the medical literature. 

## 2. Results and Discussion

### 2.1. Mobilization

Mobilization of HSC from the BM into peripheral blood (PB) is usually measured by the number of circulating CD34^+^ and/or nucleated blood cells harvested using leukapheresis [42]#. The standard mobilization agent is recombinant granulocyte colony-stimulating factor (G-CSF; filgrastim or lenograstim), an endogenous growth factor responsible for inducing granulocyte expansion and maturation in times of infection or stress [43]#. 

Within the group of chemokines (Table 1), the *CXCL12 3’UTR A* allele (rs1801157; g.44372809G>A) has shown positive correlation with mobilization in both healthy donors and patients undergoing autologous transplantation [44,45,46]. The functional consequence of this *CXCL12* (*SDF-1*) polymorphism is still unclear, but it may lead to lower protein levels [47]#. This would concur with abounding evidence on CXCR4, the CXCL12 receptor, whose blockade promotes mobilization when using plerixafor. 

As expected, several publications on plerixafor (47) were relevant, with most assessing mobilization for autologous HSCT in MM and lymphoma patients. Although an improvement may be achieved by increasing the dose [48]#, plerixafor has been demonstrated to be less efficient as a monotherapy than in combination with G-CSF [49]. Interestingly, in patients responding poorly to G-CSF (<  20 × 10^6^/L CD34+ cells in PB), pre-emptive plerixafor treatment led to a final yield equivalent to a rescue strategy administered to patients with insufficient leukapheresis [50]. Several additional studies have endorsed the use of plerixafor in autologous transplantation for diabetic patients [51] and pediatric patients [52,53,54], whereas other articles have supported its use in elderly patients and those with renal insufficiency [55,56]#. 

Two early studies also showed plerixafor to be efficient in mobilizing healthy allogeneic donors with a reasonable safety profile [57,58], and this was later reported by a phase I/II trial [59]#. Examining these varied studies, an extension of plerixafor indications is to be expected in the coming years, as are new pharmacological alternatives. Indeed, new compounds targeting CXCR4 are in development: small molecules (TG-0054 [60,61,62]) such as plerixafor, but also peptides (BL-8040 [63], (BK)T140 [64], POL6326 [65], LY2510924 [66]), or oligonucleotides (NOX-A12 [67]). All have already been tested in humans as part of phase I or early phase II clinical trials. 

Finally, although the CD34^+^ count in PB remains the most used predictor for guiding cost-efficient mobilization regimens [68]#, new biomarkers are being eagerly sought to improve individualized prescriptions. Nonetheless, the expression of CXCR4 in CD34^+^ HSC in correlation with mobilization has thus far shown discordant findings [69,70,71], and additional studies are needed.

### 2.2. Engraftment

Engraftment in humans is assessed in PB and defined by the stable recovery of blood cell counts after myeloablative conditioning and graft infusion: platelets > 50 × 10^9^/L in the absence of transfusion (platelet engraftment); or neutrophils > 500 × 10^6^/L (neutrophil engraftment) [105]#. In allogeneic HSCT, additional genetic testing for chimerism is performed to confirm the donor origin of the hematopoietic recovery [106,107]#. The absence of engraftment or the loss of donor cells after initial engraftment constitute primary and secondary graft failure (GF), respectively [108]#. In animal studies, mostly on mice, competitive repopulation assays allow for a much larger toolkit of measurements of HSC engraftment capacity [109]#. 

The use of anti-CXCR4 compounds for mobilization in the donor did not preclude engraftment in humans [61,85,94,95,110,111] or mice [112], with some studies reporting even better engraftment in mice [113,114] (Table 2). Targeting CXCR4 could also improve engraftment by vacating the hematopoietic niches in the recipient before HSCT, either via chimeric antigen receptor (CAR) T cells co-expressing CXCR4 and C-kit or via plerixafor [115,116,117]. Despite discordant results in mice [118], plerixafor administration post-HSCT in human recipients improved engraftment in one phase I/II clinical trial [119]. In this study, “mobilizing” doses of plerixafor were started from day 2 post-HSCT and continued until day 21 or neutrophil engraftment. 

Conversely, CXCR4 expression in both mice and human cells correlated positively with autologous and xeno-engraftment [120,121]. In humans, following G-CSF mobilization, CXCR4 expression showed a positive correlation with engraftment [122,123,124]. Surprisingly, here, the *CXCL12 3’UTR A* polymorphism whose occurrence had been associated with increased mobilization (see the Mobilization subsection) was associated with faster hematopoietic recovery in autologous transplant patients [125]. Indeed, if it really decreased protein expression, one would expect reduced homing of the graft CXCR4+ HSC by CXCL12-expressing stromal cells. However, more research seems warranted to define the timing of CXCR4 requirements both before and during the course of engraftment.

CXCL12-CXCR4 may also act indirectly. Prostaglandin E2 (PGE2) ex vivo treatment of murine HSC improved their BM homing and engraftment through increased expression of CXCR4 [126,127,128,129]. Similarly, inhibition of Bone Morphogenetic Protein (BMP) signaling in recipients increased CXCL12 levels and engraftment [130]. In a zebrafish model, CXCL8/CXCR1 expression by endothelial cells in the hematopoietic niche helped HSC engraftment, partly via CXCL12 upregulation [131]. 

Concerning other chemokines, high levels of interferon gamma-dependent CXCL9 [132,133] have been associated with GF in humans. In mice, knocking out (CXCR2) delayed hematopoietic recovery [134]. On the other hand, CCR1 expression marked human HSC as responsible for high levels of xeno-engraftment in mice [135]. These are some examples of the contribution of chemokines to hematopoietic-niche integrity.

There is less evidence available for other classes of GPCR. For instance, the engraftment of cells mobilized by cannabinoid receptor 2 (CB2) agonism [136] in animals or Beta-3 adrenergic receptor (B3AR) agonism [102] in humans was equivalent to those mobilized by G-CSF. Frizzled-6 (Fzd-6), a class F GPCR for Wnt protein ligands [137]#, is another potential contributor, as it was shown to be necessary for BM reconstitution beyond the homing phase [138]. A potentially clinically relevant finding is the presence of auto-antibodies activating Angiotensin 1 receptor (AT1R) in human allogeneic HSCT recipients, described in auto-immune settings [139]# and solid organ allo-rejection [140]#, and their association with decreased engraftment[141]. 

### 2.3. Sinusoidal Obstruction Syndrome (SOS)

Some early HSCT complications such as thrombotic microangiopathy and SOS are initiated by endothelial cell damage [156,157]#. SOS, formerly called veno-occlusive disease of the liver (VOD), occurs in 5–60% of HSCT patients, depending on prophylaxis and risk factors [158,159,160]# such as the underlying disease, the use of alkylating agents for conditioning, patient age, or liver disease. Sinusoidal endothelial cell damage is the key step in the pathophysiology of SOS, leading to the activation of the coagulation cascade, centrilobular thrombosis and consequent post-sinusoidal hepatic hypertension and, potentially, multiple-organ failure [157]#. Clinically, SOS is characterized by jaundice, fluid retention, painful hepatomegaly, and often thrombocytopenia refractory to transfusion [160,161]#.

Our review strategy identified no direct associations between any GPCRs and SOS occurrence or severity, yet some additional reports caught our attention. For example, recombinant thrombomodulin (rTM) is approved in Japan to treat disseminated intravascular coagulation (DIC) and has been shown to reduce SOS and the occurrence of thrombotic microangiopathy in HSCT patients [162,163]#. In two murine SOS models, one using monocrotaline (MCT) and the other using busulfan/cyclophosphamide conditioning followed by HSCT, rTM’s cytoprotective effect was demonstrated to depend on its fifth epidermal growth factor-like region (TME5) [164,165]#. A murine model of tacrolimus-induced vascular injury showed that the pro-angiogenic functions of TME5 depended on its binding to G protein-coupled receptor (GPR) 15 [165,166]#. rTM was able to mitigate aGvHD in mice in a GPR15-dependent manner [167]#. However, this GPR15 dependency has yet to be demonstrated directly for SOS in vivo. Interestingly, the oligonucleotide—defibrotide—the only FDA/European Medicines Agency (EMA)-approved drug for the treatment of SOS [168]#, was shown to increase thrombomodulin expression in humans [169]#. 

A traditional Japanese medicine called Dai-kenchu-to (DKT) was able to attenuate liver damage but not prevent the development of SOS induced by MCT [170]#. As a potential mechanism, MCT-induced CXCL1 (or CINC1) upregulation was suppressed in the DKT-treatment group, which could be a potential mechanism for explaining the associated reduction of neutrophil accumulation in the liver. 

### 2.4. Graft-Versus-Host Disease (GvHD)

#### 2.4.1. Acute GvHD

Acute GvHD (aGvHD) occurs when naïve T cells from an allogeneic donor are activated by recipient or donor antigen-presenting cells to attack recipient cells [171]#. This process is triggered by the inflammatory setting of HSCT. Once activated within lymph nodes, the alloreactive effector T cells migrate to the skin, the gastrointestinal (GI) tract, or the liver, causing further inflammation and damage [172]#. Some of the main determinants of aGvHD risk are the sources of HSCs themselves, donor–recipient HLA mismatches, the intensity of the conditioning regimen, and the absence of any GvHD prophylaxis [173]#. Immunosuppression is systematically used to prevent and treat aGvHD [174]#. Like other immune cells, T cell trafficking is regulated by myriad chemo-attractants, including chemokines. A study of the expression kinetics of a panel of chemokines and receptors in GvHD-target organs following allo-HSCT compared that expression to the histopathological changes occurring in the same organs [175]#. Characterization of the individual contributions of each chemokine/receptor would be needed to make further conclusions, but it highlights that aGvHD is a dynamic process with a complex spatiotemporal network of chemo-attractants at play. 

A number of chemokines or their receptors are associated with the development of aGvHD (Table 3). For instance, higher CCL8 levels correlated with more severe murine aGvHD [176], and CCR2 expression on CD8^+^ effector T cells was necessary for their migration to the murine gut and the liver and for the generation of aGvHD [177]. In contrast, broad inhibition of CCL2, CCL3, and CCL5 reduced murine liver aGvHD [178]. Also in mice, anti-CD3 treatment during preconditioning reduced aGvHD by limiting both CCR7^+^ dendritic cells homing to lymph nodes and CCR9^+^ effector T cells homing to aGvHD target organs without reducing GvL [179]. In humans, both a CCL5 (RANTES; Regulated on Activation, Normal T Cell Expressed and Secreted) haplotype of three polymorphisms [180] and the expression of the CX3CL1/CX3CR1 pair [181] positively correlated with the occurrence of aGvHD. Depending on the cells bearing GPCRs, other chemokine receptors can prevent aGvHD. The presence of CCR8 on regulatory T cells (Tregs) is crucial to their anti-GvHD action in mice [182], whereas Chem23R, another chemo-attractant receptor [183], prevents intestinal aGvHD in mice [183]. The *CXCL12 3’UTR A* allele previously discussed for mobilization and engraftment was here associated with reduced risk and severity of aGvHD [184], highlighting the favorable prognosis carried by this allele. The anti-CCR4 antibody, mogamulizumab, is currently approved for human use before HSCT to treat certain adult T-cell leukemias. This might accelerate subsequent aGvHD because it not only targets CCR4^+^ tumor cells but also CCR4^+^ Tregs [185,186]. Higher CCR5 and CCR9 levels were detected on children’s memory effector T cells before they developed GI aGvHD [187]. 

CCR5 is particularly interesting and is used by human immunodeficiency virus (HIV) as a co-receptor for entry into CD4^+^ T cells, thus partly explaining the genetic susceptibility to HIV infection [188]#. Maraviroc, a CCR5-antagonist, was approved in 2016 for the treatment of HIV. In the context of HSCT, the CCR5 ∆32 mutation was first associated with lower aGvHD [189,190]. Several related studies subsequently showed different subgroups of CCR5^+^/CD4^+^ T cells could be associated with intestinal aGvHD [191,192,193]. Similarly, dendritic cells expressing CCR5 could be associated with aGvHD [194,195,196], showing that CCR5 could be a chemo-attractant for several causative immune cell types. Two phase I/II trials have now tested the safety and the efficacy of CCR5 blockade using maraviroc for the prevention of aGvHD. The first trial, conducted in adults [197], proved successful and led to follow-up studies by the same group of researchers [198,199,200,201,202] as well as an ongoing phase II study (NCT01785810). Another trial [203], published in 2019, included adults and children but had inconclusive findings due to unrelated toxicities. According to its authors, CCR5 blockade could prevent lymphocyte homing but not their activation, highlighting the temporal complexity of immune activation. In mice, three studies have shown the absence of CCR5 to accelerate aGvHD [204,205,206]. Nevertheless, more recent studies demonstrated that another anti-CCR5 antibody, (PRO-140) [207], or maraviroc combined with either cyclosporine A [208,209] or CXCR3 blockade [210], could indeed prevent aGvHD in mice. CXCR3 also demonstrates a compelling case. One study showed that CXCR3-expressing Tregs could mitigate aGvHD [211], and more recent studies indicated a positive correlation between CXCR3 expression and aGvHD in mice [210,212,213,214,215,216,217] and humans [200,218]. One of these proposed CXCR3-signaling as a resistance mechanism to CCR5 blockade [200]. As for CCR6 and 7 [179,194,195,219,220,221,222,223,224] or CXCR2 and 4 [225,226,227,228,229], the evidence has been too heterogeneous and conflicting to draw any conclusions. Using combined blockade at several steps in the immune activation underlying aGvHD could create a synergy to reduce its severity. However, additional studies are needed, especially to assess the potential of abrogating the GvL effect with such an approach. 

Among adrenergic receptors, alpha-2 adrenergic receptor (A2AR) agonism [230,231] or beta-adrenergic receptor (BAR) activation under stressful conditions [232,233] was associated with lower aGvHD in mice, and so was P2Y_2_ knock-out [234]. The previously mentioned AT1R auto-antibodies were also revealed to be associated with increased aGvHD in humans [141]. Some interesting candidates have also emerged from the class of lipid mediators. The role played by the endocannabinoid system (ECS) in inflammation is now established [28]#, and the ECS was previously implicated in solid organ rejection [235,236]#. In mice, CB1/2 activation with tetrahydrocannabinol (THC) was able to mitigate aGvHD [237], whereas transplants where CB2 was knocked-down induced higher aGvHD [238]. In a human phase II trial, cannabidiol was also able to prevent aGvHD [239]#. The broad S1P_1_ agonist, fingolimod, is approved for the treatment of multiple sclerosis and works by sequestering lymphocytes in secondary lymphoid organs [240]#. A more specific agonist (CYM-5442) was shown to reduce the severity of murine aGvHD by inhibiting macrophage recruitment via a reduction of CCL2 and CCL7 expression on endothelial cells [241].

Among the other GPCR classes, complement 3/5 activator fragments receptors (C3aR/C5aR) [242] or platelet-activating factor receptor (PAFR) [243] in mice, as well as a microsatellite in human EGF, Latrophilin, and Seven Transmembrane Domain-Containing Protein 1 (ELTD1) [244], have all shown positive correlation with aGvHD. A frizzled agonist was able to rescue LGr5^+^ gastric stem cells from murine aGvHD [245], underlining the importance of each target organ’s microenvironment. Activated protein C (aPC) signaling using protease-activated receptor 2/3 (PAR 2 and 3) expanded Tregs and mitigated aGvHD in mice [246]. rTM depends on GPR15 to mitigate murine aGvHD [167], whereas human patients receiving rTM were shown to have lower CCL5 levels and aGvHD [247]. The case of GPR43 merits further discussion; it is a sensor of gut microbiota-derived metabolites, such as short-chain fatty acids (SCFAs). These metabolites limit a number of inflammatory processes via action on endothelial cells [248]# by modulating neutrophil recruitment [249]# or the CD8^+^ T cell’s effector function [250]#. In the intestine, GPR43 contributes to epithelial integrity, and GPR43 knock-out in mice was associated with the increased severity of aGvHD [251].

#### 2.4.2. Chronic GvHD

Chronic cGvHD (cGvHD) historically develops from 100 days after allogeneic HSCT, but it can nevertheless overlap with aGvHD, as it shares some initiating events, although it has different pathophysiology and clinical manifestations [257]#. Although it has not been completely elucidated, cGvHD pathogenesis involves chronic inflammation, aberrant tissue repair, and fibrosis, while the underlying immune dysregulation affects multiple cell types [258]#. The therapeutic arsenal against cGvHD is limited [259,260]#, making cGvHD the main contributor to TRM in long-term HSCT survivors [261]#. Due to its timescale, cGvHD overlaps with other chronic and/or age-related conditions, such as metabolic syndrome, chronic infections, or second primary cancers [262]#. cGvHD can affect virtually any organ but strikes the following systems in particular: skin and its appendages, mucosae, muscles and joints, and lungs. 

High levels of several CCL and CXCL chemokines [263,264,265,266,267] have shown positive correlation with cGvHD, although most of the evidence originated from a single animal study [263] (Table 4). The expression of CXCL9 [267,268,269,270,271], CXCL10 [265,266,269,270,272,273], and CXCL11 [270], as well as their common receptor, CXCR3 [269,270,271,272,273,274], correlated positively with cGvHD in humans. Some chemokine receptors (CCR1, 3, 6, 7, 9) [263,275,276,277] showed positive correlation, whereas the correlation with some others (CXCR5, CX3CR1) [278,279] was negative. The available evidence on CCR4 and CCR5 is conflicting. High CCR4/5 levels in the buccal mucosa and salivary glands have been associated with higher T cell infiltrates and cGvHD [265]. However, in another study, CCR4^+^ CD4^+^ T cells were associated with lower cGvHD [280], although the authors did not specify the subset of CD4^+^ T cells in question. Similarly, lower CCR5 on monocytes could be associated with cGvHD in joints [279]. Considering the large variety of immune cells involved in cGvHD, chemokines are naturally expected to play different roles during its course. It is still interesting to note that, as for aGvHD, CB2 knock-out is associated with more severe cGvHD. To date, evidence for the other GPCRs [Prostaglandin D2 receptor (PGD2R), AT1R, smoothened (SMO), CB2] is either conflicting or based on single studies. 

### 2.5. Lung Toxicity

Pulmonary complications following HSCT are a cause of morbidity and mortality. They arise from infections, iatrogenic fluid overload, idiopathic pneumonia syndrome (IPS), and as a consequence of renal or cardiac failure or cGvHD [285]#. IPS is an early complication of allogeneic HSCT that encompasses a spectrum of clinical presentations arising from acute, widespread alveolar injury [286]#. The type and the intensity of conditioning medication, especially cyclophosphamide, and the activation and the migration of donor T cells are important contributors to that injury [287,288]#. Various cellular and soluble inflammatory mediators are thought to play a role in the development of IPS [286,289]#. Our systematized search of the literature found several animal studies reporting associations between chemokines and/or receptors and the occurrence of IPS (Table 5). CXCL 9 and 10, and their receptor, CXCR3 [290], in addition to CCL5 (RANTES) [178,291], showed positive correlation with IPS. For CCL2 [178,292,293] and CCL3 [178,294], the evidence was more conflicting. As with GvHD, specific chemokines probably correlate with distinct immune cell functions during the course of IPS [286]#. No reports of associations were found for any other functional classes of GPCR. It is interesting to note that no new studies on this have been published in the last ten years.

### 2.6. Treatment-Related Mortality (TRM)

TRM comprises deaths not due to the underlying disease. In cases of malignant diagnoses, this means death not due to a relapse of the disease, also sometimes called non-relapse mortality (NRM) [295]#. GvHD, VOD, lung toxicity, and infections due to HSCT-related immunosuppression are important causes of TRM [285]#. Mortality rates are tightly linked to responses to the initial treatments for each one of those complications; refractory aGvHD, for instance, is fatal in up to 80% of cases [285]#. Animal transplantation models do not usually recapitulate the underlying diseases for which human HSCTs are indicated, thus defining TRM seems futile in animal studies. In contrast, in human studies, two human polymorphisms in CCL2 (rs1024610, NG_012123.1:g.2936T>A) [296] and CXCL10 (rs3921, NM_001565.3:c.*140G>C) [297] have been associated with increased and decreased TRM, respectively (Table 6). The study on CCL2 found no significant associations between the variant and aGvHD, suggesting that another TRT could be the cause of the observed TRM. In contrast, in the study on the CXCL10 variant, lower TRM was associated with lower organ failure. CXCL9 was part of a four-biomarker panel associated with TRM [268]. High CCR5 expression in recipient T cells increased TRM [198,254], whereas a CD4^+^ CCR5^+^ cell population was associated with higher TRM [191]. In another study, CCR7^+^ CD4^+^ T cells were associated with death from cGvHD [298]. No reports of associations were found for any other functional class of GPCR.

## 3. Methods

### 3.1. Systematized Search

We used MEDLINE (www.ncbi.nlm.nih.gov/pubmed/) and EMBASE (www.embase.com/) databases to carry out a systematized review [299]# of articles published in English up to 4 March 2019 (see [App secA4-ijms-20-03889]). The search extended to in vivo models and human interventional and non-interventional studies (see [App secA3-ijms-20-03889]). The following HSCT outcomes were selected: mobilization, engraftment, SOS, acute GvHD, chronic GvHD, lung toxicity, and TRM. The rationale for this selection and the measurement methods are explained in the [App secA8-ijms-20-03889]. Due to the advances in research on plerixafor and the existence of another recent systematic report reviewing its use for its approved indications [41]#, we restricted our search to human studies where mobilization was the measured outcome. The Preferred Reporting Items for Systematic Reviews and Meta-Analyses (PRISMA; www.prisma-statement.org/) guidelines were followed to ensure a systematized search, although some of the requirements were not applicable due to the quite inclusive selection criteria used [300]#, as explained in the [Sec secA6-ijms-20-03889], [Sec secA7-ijms-20-03889], [Sec secA8-ijms-20-03889], [Sec secA9-ijms-20-03889], [Sec secA10-ijms-20-03889] and [Sec secA11-ijms-20-03889]. The selection and data collection processes are described in the [Sec secA6-ijms-20-03889] and [Sec secA7-ijms-20-03889] as well. The search workflow and its output are reported in Figure 1 below, whereas Figure 2 details the number of published articles per year. In the Results and Discussion section, we report on and discuss the results of the search. 

### 3.2. Reporting of the Results

In the Results and Discussion section, each table lists the GPCRs, GPCR ligands, or related proteins whose expression/activity was reported to correlate with the HSCT outcome under consideration, as well as the corresponding reference(s) from the systematized search. Whenever an increase in gene/protein expression or activity of a GPCR, a GPCR ligand, or a related protein was associated with an increase in the incidence/level/severity of the outcome under consideration, the correlation is described as positive (+). The same applies whenever a decrease in a GPCR expression/activity was associated with a decrease in the outcome. Conversely, whenever an increase or a decrease in the GPCR expression/activity was associated with a decrease or, respectively, an increase in the outcome, the correlation is negative (−). Whenever there was no association between a GPCR expression/activity and the outcome, the correlation is null (0). As for polymorphisms (identified as “haplotype”, “microsatellite”, or by the variant number), their presence can correlate either positively (+) or negatively (−) with the outcome, yet their effect on protein level/function is not necessarily known. GPCRs or their ligands are grouped according to functional classes: chemokines [C-C ligand/receptor (CCL/R), C-X-C ligand/receptor (CXCL/R), C-X3-C ligand/receptor (CX3CL/R) blue], adrenergic receptors (orange), lipid mediators/receptors (green), and “others” (gray). To introduce topics or to enrich the discussion, we considered additional studies, which were not selected by the research query and/or criteria, as well as reviews. These references, along with those cited in the introduction, are specifically identified (#).

## 4. Conclusions and Perspectives

This systematized review reports on a significant number of GPCRs showing consistent associations with mobilization and engraftment and for which research has moved on to more advanced stages. Although there is some evidence that GPCRs play a role in SOS, GvHD, lung toxicity, and TRM in HSCT settings, there is a flagrant paucity of clinical associations. For several target GPCRs, the evidence is lacking or conflicting. In contrast, chemokines and their receptors make promising potential targets/biomarkers, as there are numerous potential candidates in various settings. Despite the difficulties in isolating the contributions of individual GPCRs, research has made significant progress for several of them. Targeting CXCR4 for mobilization has proven its utility, with the marketing authorization of plerixafor coming in 2008. Further work is needed to extend plerixafor’s indications, and the new anti-CXCR compounds in development could offer interesting pharmacological alternatives. The timing of CXCR4′s role during engraftment remains unclear, but CXCR4 blocking during mobilization does not seem to prevent engraftment, and CXCR4 could be manipulated so that it vacates the recipient niche or stimulates engraftment. No direct link between a GPCR and SOS has been consistently demonstrated in vivo. As for aGvHD, CCR5 blockade, such as with the anti-HIV drug, maraviroc, is on track to become a therapeutic option for its prophylaxis. Combined or alternated blockade using CXCR3 and CCR5 might bring further benefits. Activating cannabinoid receptors could be another prospect. GPR43 also merits further investigation as the importance of the gut’s microbiota in inflammatory processes is increasingly recognized. 

It seems that research on GPCRs in the context of cGvHD is less advanced than in that of aGVHD. The current state of knowledge involves multiple chemokines but is either based on single studies or reports with conflicting findings. Studies on lung toxicity and IPS were scarce, and no relevant contribution to this field has been made in the last ten years. For both cGvHD and IPS, a better understanding of the molecular pathogenesis will probably be required before any useful biomarkers are revealed. As for TRM, it is often multifactorial and may thus prove more challenging to associate death with a single biomarker than individual or even combined toxicities. The absence of an assumed common toxicity-related pathway may explain the paucity of studies revealed by our literature search strategy. 

The methodology used in the present paper strived to follow the PRISMA protocols, which, due to the nature of the search performed, could not be followed strictly. However, given the broad range of GPCRs, using the PRISMA methodology helped the authors to guide their search, resulting in a systematized review [299]. Because our search included both pre-clinical and clinical studies, quality, precision, and developmental stage of the evidence was inevitably heterogeneous and could not be reported or summarized using quantitative measures. Despite our best efforts to cover all GPCRs, certain reports that we judged significant enough to mention were missed by the search strategy. Nevertheless, the structure of this article allowed the authors to include such papers in the Results and Discussion section in order to properly cover the subject. Also, time and human resources limitations did not allow for quality or bias assessment by multiple unbiased reviewers, as should be expected from a completely systematic review. Regardless of the limitations to our systematized approach, it did allow for comprehensive scope and was meant to inform scientists and clinicians of the latest developments in a field that is (re-)gaining momentum.

## Figures and Tables

**Figure 1 ijms-20-03889-f001:**
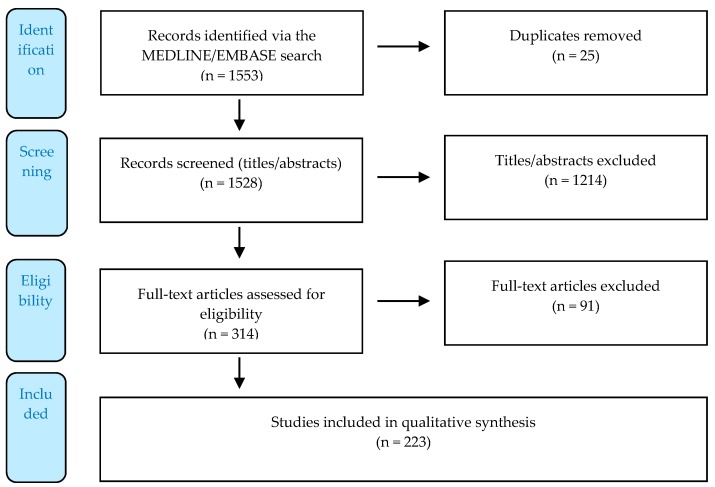
Flow diagram displaying the number of records identified, included, and excluded via a systematized literature review following Preferred Reporting Items for Systematic Reviews and Meta-Analyses (PRISMA) guidelines [300]#.

**Figure 2 ijms-20-03889-f002:**
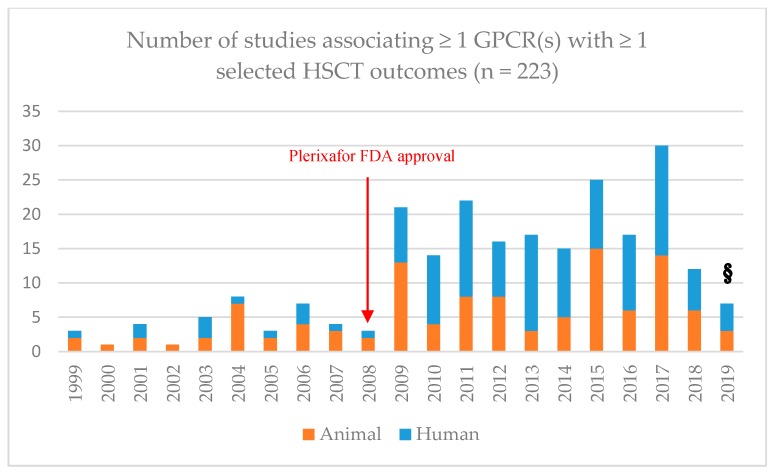
Number of animal (orange) or human (blue) studies per year reporting any association between at least one G protein-coupled receptor (GPCR) and at least one of the selected hematopoietic stem cell transplantation (HSCT) outcomes: mobilization, engraftment, treatment-related toxicities (TRTs) [veno-occlusive disease (VOD), acute graft-versus-host disease (aGvHD), chronic graft-versus-host disease (cGvHD), lung toxicity], and transplantation- or treatment-related mortality (TRM). N = 223. These records were selected via two rounds of systematized screening for eligibility/exclusion criteria (see the [App secA6-ijms-20-03889]). §: up to 4 March 2019.

**Table 1 ijms-20-03889-t001:** Mobilization in human (H) studies. Mobilization is measured by the number of circulating CD34^+^ (HSC) and/or nucleated blood cells harvested using leukapheresis. See the Methods section regarding the reporting of results (Section 3.2).

Mobilization			
	Studies	Correlation with outcome	References
C-X-C ligand 8 (CXCL8)	H	+	[72]
C-X-C ligand 12 (CXCL12)	H	0	[73]
	H (rs1801157)	+	[44,45,46]
C-X-C receptor 4 (CXCR4)	H	−	[49,50,51,52,53,54,57,58,60,61,62,63,64,65,66,67,69,70,71,74,75,76,77,78,79,80,81,82,83,84,85,86,87,88,89,90,91,92,93,94,95,96,97,98,99,100,101]
Beta-3 adrenergic receptor (B3AR)	H	+	[102]
Protease-activated receptor 1 (PAR1)	H	+	[103]
Relaxin/insulin-like family peptide receptor 4 (RXFP4)	H	+	[104]

**Table 2 ijms-20-03889-t002:** Engraftment in animal (A) or human (H) studies. In humans, engraftment is measured by either the time to platelet/neutrophil recovery, chimerism, or the absence of graft failure. In animals, genetic manipulation allows for various measures of engraftment. See the Methods section regarding the reporting of results (Section 3.2).

Engraftment			
	Studies	Correlation with Outcome	References
C-C ligand 15 (CCL15)	A	+	[142]
C-X-C ligand 9 (CXCL9)	H	−	[132,133]
C-X-C ligand 12 (CXCL12)	H (rs1801157)	+	[125]
	A	+	[130]
C-C receptor 1 (CCR1)	A	+	[135]
C-X-C receptor 1 (CXCR1)	A	+	[131]
C-X-C receptor 2 (CXCR2)	A	+	[134,143]
C-X-C receptor 4 (CXCR4)	A, H	+	[120,121,144,145,146,147] (A), [122,123,124,148] (H)
		0	[112,118] (A), [61,85,94,95,110,111] (H)
		−	[113,114,115,116,117,149] (A), [119] (H)
Gai-coupled chemokine receptors (Pertussis toxin)	A	0	[150]
Angiotensin 1 receptor (AT1R)	H	−	[141]
Beta-3 adrenergic receptor (B3AR)	H	0	[102]
Cannabinoid receptors 1/2 (CB1/CB2)	A	0	[136]
Prostaglandin E2 (PGE2)	A	+	[126,127,128,129]
Prostaglandin I2 (PGI2)	A	+	[151,152]
Sphingosine-1-phosphate receptor 3 (S1PR3)	A	−	[153]
Calcium receptor (CaR)	A	+	[154]
Frizzled-6 (Fzd-6)	A	+	[138]
GPCR-associated sorting protein 2 (Gprasp2)/Armadillo repeat-containing X-linked protein 1 (Armcx1)	A	−	[155]

**Table 3 ijms-20-03889-t003:** Acute GvHD occurrence/severity in animal (A) or human (H) studies. See the Methods section regarding the reporting of results (Section 3.2).

aGvHD			
	Studies	Correlation with Outcome	References
C-C ligand 2 (CCL2)	A	+	[178]
C-C ligand 3 (CCL3)	A	+	[178]
C-C ligand 5 (CCL5; RANTES)	A	+	[178]
	H	+	[180]
	H (haplotype)	+	[247]
C-C ligand 8 (CCL8)	A	+	[176]
C-X-C ligand 10 (CXCL10)	A	+	[252]
C-X-C ligand 12 (CXCL 12)	H (rs1801157)	+	[184]
C-X-C ligand 13 (CXCL 13)	A	+	[253]
C-X3-C ligand 1 (CX3CL1)	A	+	[181]
C-C receptor 1 (CCR1)	A	+	[175]
C-C receptor 2 (CCR2)	A	+	[175,177]
C-C receptor 4 (CCR4)	H	−	[185,186]
C-C receptor 5 (CCR5)	A, H	+	[207,208,209,210](A), [187,189,190,191,192,193,194,195,196,197,198,199,200,201,202,254] (H)
	H	0	[203]
	A	−	[204,205,206]
C-C receptor 6 (CCR6)	A, H	+/−	[223] (A, −)/[219] (H, +)
C-C receptor 7 (CCR7)	H	+	[179,194,195]
	A, H	−	[220,221] (A), [222,223,224] (H)
C-C receptor 9 (CCR9)	H	+	[179,187]
C-C receptor 8 (CCR8)	A	−	[182]
Chemerin receptor 23 (Chem23R)	A	−	[183]
C-X-C receptor 2 (CXCR2)	A	+/−	[226](+)/[225] (−)
C-X-C receptor 3 (CXCR3)	A, H	+	[175,210,212,213,214,215,216,217](A), [200,218] (H)
	A	0	[255]
	A	−	[211,256]
C-X-C receptor 4 (CXCR4)	A, H	+	[225] (A), [227] (H)
	A	−	[228,229] (A)
Alpha-2 adrenergic receptor (A2AR)	A	−	[230,231]
Angiotensin 1 receptor (AT1R)	H	+	[141]
Beta-adrenergic receptor (BAR)	A	−	[232,233]
P2Y purinoreceptor 2 (P2Y_2_)	A	+	[234]
Cannabinoid receptor 1 and 2 (CB1/CB2)	A	−	[237]
Cannabinoid receptor 2 (CB2)	A	−	[238]
Sphingosine-1-Phosphate 1 (S1P1)	A	−	[241]
Complement 3/5 activator fragments receptors (C3aR/C5aR)	A	+	[242]
EGF, Latrophilin and Seven Transmembrane Domain-Containing Protein 1 (ELTD1)	H (microsatellite)	+	[244]
G Protein-Coupled Receptor 15 (GPR15)	A	−	[167]
G Protein-Coupled Receptor 43 (GPR43)	A	−	[251]
Leucine-rich repeat-containing G protein-coupled receptor 5 (LGR5)	A	−	[245]
Platelet-activating factor receptor (PAFR)	A	+	[243]
Protease-activated receptor 2/3 (PAR2/3)	A	−	[246]

**Table 4 ijms-20-03889-t004:** Chronic GvHD occurrence/severity in animal (A) or human (H)studies. See the Methods section regarding the reporting of results (Section 3.2).

cGvHD			
	Studies	Effect	References
C-C ligand 2 (CCL2)	A	0	[281]
C-C ligand 3 (CCL3)	H	+	[266]
C-C ligand 5 (CCL5)	A	+	[264]
C-C ligand 6 (CCL6)	A	+	[263]
C-C ligand 7 (CCL7)	A	+	[263]
C-C ligand 8 (CCL8)	A	+	[263]
C-C ligand 9 (CCL9)	A	+	[263]
C-C ligand 11 (CCL11)	A	+	[263]
C-C ligand 19 (CCL19)	A	+	[263]
C-C ligand 22 (CCL22)	H	+	[265]
C-X-C ligand 2 (CXCL2)	A	+	[263]
C-X-C ligand 8 (CXCL8)	H	+	[266]
C-X-C ligand 9 (CXCL9)	A, H	+	[263] (A), [267,268,269,270,271] (H)
C-X-C ligand 10 (CXCL10)	A, H	+	[263] (A), [265,266,269,270,272,273] (H)
C-X-C ligand 11 (CXCL11)	H	+	[270]
C-X-C ligand 12 (CXCL12)	A	+	[263]
C-C receptor 1 (CCR1)	A	+	[263]
C-C receptor 2 (CCR2)	A	0	[281]
C-C receptor 3 (CCR3)	H	+	[276]
C-C receptor 4 (CCR4)	H	+/−	[265] (+)/[280] (−)
C-C receptor 5 (CCR5)	H	+/−	[265] (+)/[279] (−)
C-C receptor 6 (CCR6)	H	+	[275]
C-C receptor 7 (CCR7)	H	+	[282]
C-C receptor 9 (CCR9)	H	+	[277]
C-X-C receptor 3 (CXCR3)	H	+	[269,270,271,272,273,274]
C-X3-C receptor 1 (CX3CR1)	H	−	[279]
C-X-C receptor 5 (CXCR5)	A	−	[278]
Angiotensin 1 receptor (AT1R)	H	+	[283]
Cannabinoid receptor 2 (CB2)	A	−	[238]
Prostaglandin D2 receptor (PGD2R; CRTH2)	H	+/−	[276] (+)/[280] (−)
Smoothened (SMO)	A	+	[284]

**Table 5 ijms-20-03889-t005:** Lung toxicity occurrence/severity in animals (A) or humans (H). See the Methods section regarding the reporting of results (Section 3.2).

Lung Toxicity			
	Studies	Effect	References
C-C ligand 2 (CCL2)	A	+/0	[178,292] (+)/[293](0)
C-C ligand 3 (CCL3)	A	+/−	[178] (+)/[294] (−)
C-C ligand 5 (CCL5) or Regulated on Activation, Normal T Cell Expressed and Secreted (RANTES)	A	+	[178,291]
C-X-C ligand 9 (CXCL9)	A	+	[290]
C-X-C ligand 10 (CXCL10)	A	+	[290]
C-C receptor 2 (CCR2)	A	0	[293]
C-X-C receptor 3 (CXCR3)	A	+	[290]

**Table 6 ijms-20-03889-t006:** Treatment-related mortality (TRM) in humans (H). See the Methods section regarding the reporting of results (Section 3.2).

Treatment-Related Mortality			
	Studies	Effect	References
C-C ligand 2 (CCL2)	H (rs1024610)	+	[296]
C-X-C ligand 9 (CXCL9)	H	+	[268]
C-X-C ligand 10 (CXCL10)	H (rs3921)	−	[297]
C-C receptor 5 (CCR5)	H	+	[191,198,254]
C-C receptor 7 (CCR7)	H	+	[298]

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
