# Peer review of "The Biological and Clinical Relevance of G Protein-Coupled Receptors to the Outcomes of Hematopoietic Stem Cell Transplantation: A Systematized Review"

_ijms, 2019, doi:10.3390/ijms20163889_

Round 1

Reviewer 1 Report

  1. The manuscript deals with an interesting topic but is too unfocused.
  2. First of all, considering the rapid progress in the relevant field, it is very doubtful to discuss the review articles which were published more than 10 years ago. The authors should restrict their discussion to the papers which were published less than 10 years ago, unless a review article is of paramount importance from the authors’ viewpoint.
  3. Moreover, the authors did not discuss the present status of hematopoietic stem cell transplantation, particularly the problems which requires urgent solutions. The authors should discuss these points definitively, in order to help understand the significance of each review article.
  4. Moreover, the authors did summarize each review article without evaluating the content of each paper and consequently, the review article is just a constellation of the incomplete summary of the previous review articles. The authors should evaluate each review article by themselves, to provide an overview of the present status of researches on G protein-coupled receptors in hematopoietic stem cell transplantation.
  5. Finally, the authors should provide information on the expression pattern of each G protein-coupled receptor among bone marrow cells in detail.

Author Response

Thank you for your comments. We appreciate every opportunity to improve our manuscript. To structure our response, we have taken the liberty to split your comments into different points we would like to address separately.

1. We agree with you that the topic of the review is quite broad. It is the result of a deliberate choice. We described its rationale in the Conclusion and have now clarified it in the Appendix A2 as follows.
Conclusion: “The methodology used in the present paper strived to follow the PRISMA protocols, which, due to the nature of the search performed, could not be followed strictly. However, given the broad range of GPCRs, using the PRISMA methodology helped the authors to guide their search, resulting in a systematized review [42]. Because our search included both pre-clinical and clinical studies, the quality, precision, and developmental stage of the evidence was inevitably heterogeneous and could not be reported or summarized using quantitative measures.”
Appendix A2: “Rationale and objectives: The general objective was to genuinely report on the state of knowledge regarding the identification and targeting of GPCRs in the management of hematopoietic stem cell transplantation (HSCT) outcomes. Hence, we used an inclusive approach in selecting the types of studies under consideration.”
As explained in the Appendix A1 and now highlighted in section 1.3 (lines 84-86), we searched for any existing comprehensive review covering the field of associations between HSCT outcomes and GPCR family members, however at the time of the manuscript preparation we could not find any, except regarding the use of Plerixafor in the specific indications for which it is currently approved. Yet, we came across exciting progress in (pre-)clinical studies (mainly trials targeting CCR5 to prevent aGvHD or new anti-CXCR4 compounds). It is precisely to address this gap that we chose to carry out an inclusive (see the Appendix A2) but systematized (see the Appendix A1) survey of the field we believe should be informative for clinicians, but also biologists and pharmacists. Furthermore, as the first review, to the best of our knowledge, to address the association between GPCRs and these HSCT outcomes, this manuscript could guide future research and further reviews focused on one particular HSCT outcome in association with one or a specific group of GPCRs.
2. As explained in the Appendix A3.1, reviews were excluded from the systematized search (“Conference abstracts were included, whereas reviews, editorials, and case reports were excluded.”).
We did use some reviews in the introduction, as well as in the results section either to introduce topics or to enrich the discussion, as explained in the ‘Methods’ section (lines 111-114) as follows “To introduce topics or to enrich the discussion, we considered additional studies, which were not selected by the research query and/or criteria, as well as reviews. These references, along with those cited in the introduction, are specifically identified (#).”.
Some of these reviews are more than 10 years old, but they offer either an historical perspective (e.g. ref [1]) or introduce clinical (e.g. refs [6], [7], [264], [290]) and biological (e.g. refs [12], [32], [111], [264], [289]) concepts that we found relevant.
As for (pre-)clinical studies older than 10 years, you are making an interesting point. We presented our point of view in the methodology. As we could not find any existing comprehensive review (see point 1),
we decided against setting any filter based on publication date in order to deliver a genuine assessment on the state of the field. To clarify this point, we have amended the Appendix A3.1, as follows “As there was no existing comprehensive review of the field to start from (see A1), we did not set any anterior limit on the time of publication. We ran the search for the last time on March 4th, 2019.”).
We believe that the fact that for some HSCT outcomes, such as lung toxicity, no more recent article was published, is quite informative and acknowledges the significant differences in the level of molecular understanding of each outcome.
3. As the ‘molecular pharmacology’ section of the International Journal of Molecular Sciences is not specialized in hematopoietic stem cell transplantation, we shortly introduced clinical topics (section 1.3) only to allow the non-clinicians to understand the challenges posed by mobilization, engraftment, TRTs (VOD, GVHD, lung toxicity), or TRM and how these HSCT-related outcomes could be molecularly related to GPCRs. These outcomes were selected (as now stated in the Abstract, line 22). The selection was based on the latest European Society for Blood and Marrow Transplantation (EBMT) guidelines (EBMT handbook, 2019), where these outcomes were identified as being the most common or most important
issues in the field of hematopoietic stem cell transplantation at the present time. We have clarified this in the Appendix A8, lines 556-558.
4. As stated in Point 2, we would like to highlight here again that reviews were not included in the systematized search and were used only for the enrichment of the discussion.
As for (pre-)clinical studies, we explained in the Appendices A6 to A11 and in the Conclusion (“The methodology used in the present paper strived to follow the PRISMA protocols, which, due to the nature of the search performed, could not be followed strictly. […] Because our search included both pre-clinical and clinical studies, the quality, precision, and developmental stage of the evidence was inevitably heterogeneous and could not be reported or summarized using quantitative measures. […] Also, time and human resources limitations did not allow for quality or bias assessment by multiple unbiased reviewers, as should be expected from a completely systematic review. Regardless of the limitations to our systematized approach, it did allow for comprehensive scope and was meant to inform scientists and clinicians of the latest developments in a field that is (re-)gaining momentum.”) that we did not have the resources to abide by all PRISMA requirements, such as objectively scoring the quality of each study. We are aware that this is an important shortcoming, as acknowledged in the manuscript. Yet, we believe it does not prevent this systematized survey to capture the trends and progress achieved so far in the field.
In addition, to ensure our review is not “a constellation of the incomplete summary of the previous review articles”, we have checked for similarities with existing articles and reviews using University of Geneva access to the web-based software “compilatio.net”. The only similarities we have found were based on descriptions of abbreviations (e.g. PRISMA) and protein names (e.g. RANTES). Should you request the report, we could provide it.
5. In the case of the chemokine system, we decided against explaining in details the specific location, signaling and function of each molecule, as we believe this is already thoroughly covered in the literature, as for example in this article from Charo et al. (NEJM, 2006; ref [31]) we referred to in the text (line 69). In addition, we believe it would have shifted away the focus from the main topic of this review, which is the association between the expression/activity of certain GPCRs on one hand and selected HSCT outcomes. As for other GPCRs, data on tissue expression patterns can be found on public databases such as The Human Protein Atlas or GTEx. Unfortunately, it originates from very heterogeneous sources and cell-type specific expression patterns are often not known. Extracting useful information in a non-biased manner would require a significant amount of work. We think this is beyond the scope of this survey we meant as comprehensive, yet not focused on a specific outcome or GPCR.
We have now clarified our position in section 1.2 (lines 72-74), as follows: “Discussing time and tissue-specific expression patterns of each GPCR is beyond the scope of this review, but this type of information can be found in part in the references above or in public databases such as The Human Protein Atlas or GTEx.”
language and grammar editing. The manuscript had been English-edited by a professional (see attachment), as now stated in the Acknowledgement section (lines 440-441).
We hope our response has addressed your comments satisfyingly. Should there be any additional comment/issue, we would be happy to address them.
Thank you very much for helping us improve our manuscript.

Reviewer 2 Report

In this review, Golay et al. carry out a systematic survey of the literature to identify the clinical impact of expression and targeting of GPCRs in mobilizing hematopoietic stem cells in transplant settings. The review is fairly comprehensive and the categorization of the literature in tabular format is helpful to the reader. While the review is clear for most parts, the following sections/ titles should be reworked:

1. The title of every table starts with “GPCRs, ligands, or related proteins, whose expression or manipulation is associated with…” However, it is unclear what if they mean increased or decreased expression of the indicated proteins. In addition, does the manipulation result in inhibition or activation of the indicated ligands/receptors? This information is critical for the audience to understand the impact of a chemokine or its receptor on mobilization and should be incorporated in all the tables. The authors may also consider changing the titles of some tables to break the monotony.

2. The authors should systematically review their sentences and clarify what they mean when they say “expression of X correlated with increased/ decreased engraftment”. Is this increased expression? One example is the sentence on CXCR4, line 175

3. The manuscript requires editing for language and grammar.

Author Response

Thank you for your comments. We appreciate every opportunity to improve our manuscript. To structure our response, we have taken the liberty to split your comments into different points we would like to address separately.

  1. Thank you for your suggestion. We completely agree and have changed the “Methods” section as follows:
    “Tables 1-6 will list GPCR, GPCR ligand or related protein whose expression/activity correlates with each of the following outcomes: mobilization, engraftment, SOS, acute GvHD, chronic GvHD, lung toxicity, and TRM. Whenever an increase in gene/protein expression or activity of a GPCR, a GPCR ligand or a related protein was associated with an increase in the incidence/level/severity of the outcome under consideration, the correlation is described as positive (+). The same applies whenever a decrease in a GPCR expression/activity was associated with a decrease in the outcome. Conversely, whenever an increase or decrease in the GPCR expression/activity was associated with a decrease, respectively an increase in the outcome, the correlation is negative (-). Whenever there is no association between a GPCR expression/activity and the outcome, the correlation is null (0). As for polymorphisms (identified as ‘haplotype’, ‘microsatellite’ or by the variant number), their presence can correlate either positively (+) or negatively (-) with the outcome, yet their effect on protein level/function is not necessarily known. GPCRs or their ligands are grouped according to functional classes: chemokines (CCL/R, CXCL/R, blue), adrenergic (orange), lipid mediators (green) and “others” (gray)”. We have simplified the tables subtitles accordingly and used signs (+, - or =) instead of arrows to describe the direction of correlations.
  2. Here, we also agree and have used more specific titles for tables 1-6.
  3. Thank you for providing this compelling example. We have gone through the manuscript again and hopefully fixed any similar ambiguity (lines 144, 188-191, 264-267, 294-295, 316, 338, 340-343, 363-364).
    The manuscript had been English-edited by a professional (see attachment), as now stated in the Acknowledgement section (lines 440-441).

We hope our response has addressed your comments satisfyingly. Should there be any additional comment/issue, we would be happy to address them.
Thank you very much for helping us improve our manuscript.

Round 2

Reviewer 1 Report

The authors modified the manuscript in a satisfatory manner in response to the comments.